# Platelet-Rich Fibrin Extract: A Promising Fetal Bovine Serum Alternative in Explant Cultures of Human Periosteal Sheets for Regenerative Therapy

**DOI:** 10.3390/ijms20051053

**Published:** 2019-02-28

**Authors:** Tomoyuki Kawase, Masaki Nagata, Kazuhiro Okuda, Takashi Ushiki, Yoko Fujimoto, Mari Watanabe, Akira Ito, Koh Nakata

**Affiliations:** 1Division of Oral Bioengineering, Institute of Medicine and Dentistry, Niigata University, Niigata 951-8514, Japan; 2Division of Oral Surgery, Institute of Medicine and Dentistry, Niigata University, Niigata 951-8514, Japan; nagatam@dent.niigata-u.ac.jp; 3Division of Periodontology, Institute of Medicine and Dentistry, Niigata University, Niigata 951-8514, Japan; okuda@dent.niigata-u.ac.jp; 4Bioscience Medical Research Center, Niigata University Medical and Dental Hospital, Niigata 951-8520, Japan; tushiki@med.niigata-u.ac.jp (T.U.); yfujimoto@med.niigata-u.ac.jp (Y.F.); mwatanabe@med.niigata-u.ac.jp (M.W.); radical@med.niigata-u.ac.jp (K.N.); 5Kohjin Bio Co., Ltd., Sakado 350-0214, Japan; a.ito@kohjin-bio.co.jp

**Keywords:** periosteal sheet, platelet-rich fibrin, growth, differentiation, bone grafting material

## Abstract

In 2004, we developed autologous periosteal sheets for the treatment of periodontal bone defects. This regenerative therapy has successfully regenerated periodontal bone and augmented alveolar ridge for implant placement. However, the necessity for 6-week culture is a limitation. Here, we examined the applicability of a human platelet-rich fibrin extract (PRFext) as an alternative to fetal bovine serum (FBS) for the explant culture of periosteal sheets in a novel culture medium (MSC-PCM) originally developed for maintaining mesenchymal stem cells. Small periosteum tissue segments were expanded in MSC-PCM + 2% PRFext for 4 weeks, and the resulting periosteal sheets were compared with those prepared by the conventional method using Medium199 + 10% FBS for their growth rate, cell multilayer formation, alkaline phosphatase (ALP) activity, and surface antigen expression (CD73, CD90, and CD105). Periosteal sheets grew faster in the novel culture medium than in the conventional medium. However, assessment of cell shape and ALP activity revealed that the periosteal cells growing in the novel medium were relatively immature. These findings suggest that the novel culture medium featuring PRFext offers advantages by shortening the culture period and excluding possible risks associated with xeno-factors without negatively altering the activity of periosteal sheets.

## 1. Introduction

Abundant growth factors and cytokines stored in platelet granules are released from activated platelets in response to tissue injury. These soluble factors are involved in wound healing and tissue repair [1]. In the 1990s, this essential role of platelets was exploited for regenerative therapy [2] and since then, therapies using platelet concentrates have been widely applied in various fields of regenerative medicine. In parallel with or even a little ahead of this therapeutic strategy, platelet lysates (PLs) have been used as a substitute for fetal bovine serum (FBS) [3] for in vitro cell expansion to reproducibly maintain cell proliferation [1]. Finding a possible alternative to FBS was strongly motivated by two major reasons: (1) limitation of the variability of FBS owing to the increased demands and decreased production ability; and (2) wide variability between batches that may affect end-product reproducibility, risks of pathogen contaminations, and ethical issues [1]. The quality of PLs also varied by source; however, shortage and risks of unexpected contamination could be avoided with the use of autologous platelets.

We have previously demonstrated a regenerative therapy with autologous periosteal sheets exhibiting osteogenic properties [4] for alveolar bone regeneration in more than 120 clinical cases [5,6,7] over the past 14 years on the basis of the evidence that osteogenicity, as well as osteoinductivity and osteoconductivity, are maintained in this grafting material [4,8]. Periosteal sheets are routinely expanded in vitro from small segments of alveolar periosteal tissues in the conventional medium supplemented with 10% FBS. Although no adverse events related to xeno-factors have been observed as a result of extensive washing with phosphate-buffered saline (PBS) prior to implantation, other aforementioned concerns, such as availability and efficacy of FBS, still pose difficulties. Furthermore, the requirement of a 6-week expansion period reduces the operational efficiency of cell-processing facilities, thereby increasing the economic burden. Therefore, we aimed to develop a xeno-free culture medium that may significantly shorten the period of expansion.

In a preceding study, we modified a chemically defined novel culture medium originally developed for the maintenance of mesenchymal stem cells suitable for human adult periosteal cells. This was accomplished by the addition of basic fibroblast growth factor (bFGF), platelet-derived growth factor (PDGF), and dexamethasone. Additionally, we adopted the extract of platelet-rich fibrin (PRFext) prepared from human peripheral blood samples to replace FBS. As the expansion of periosteal sheets necessitates only a limited amount of FBS replacement and as this supplement should be prepared in-house, we chose a more convenient way to obtain platelets and plasma instead of using the protocol of PL preparation. We confirmed that this novel complete medium facilitated the growth of periosteal sheets without causing genetic instability, as evident from karyotype testing. To test compatibility, we compared cell growth and fundamental characteristics of periosteal sheets prepared using the conventional culture medium (Medium199 + 10% FBS) and the newly modified stem cell medium supplemented with 2% PRFext.

## 2. Results

### 2.1. Growth of Periosteal Sheets

Figure 1 shows the onset of cell outgrowth, which indicates the days required for the migration of the first cell out of the original periosteum tissue segments. Some minor differences were reported depending on individual samples; however, no statistical difference was observed between groups. Cell outgrowth commonly occurred at 6–10 days of culture on average.

Figure 2 shows the photomicrographs of the periosteal cells that migrated out from the isolated periosteum tissue segments. The cell density was maximum in the central region in cultures with MSC-PCM + 2% PRFext (C), while the lowest density was observed in the cultures with conventional Medium199 + 10% FBS (A). Differences in cell shape were observed in the peripheral region. The majority of periosteal cells showed a typical spindle shape in the conventional medium, while their shape was relatively branched in type, indicative of their immature phenotype [9,10,11]. These findings are consistent with the results observed with MesenPRO-RS medium [8].

Figure 3 shows the growth curves of periosteal sheets. Some individual differences were observed; however, overall these data indicate that MSC-PCM + 2% PRFext was the most effective of all media. MSC-PCM + 4% FBS was equal or less effective than MSC-PCM + 2% PRFext, while the conventional medium delayed the growth of periosteal sheets.

### 2.2. Phenotype of Periosteal Sheets

Figure 4 shows alkaline phosphatase (ALP) activity, a representative phenotypic marker of differentiated osteoblasts, in fixed periosteal sheets. Safranin-O staining indicated the size of individual samples. As the cell multilayer formation varied with different types of media, it is difficult to compare ALP activity among groups.

Figure 5 shows cell multilayers and calcium deposit formation in the sagittal section of periosteal sheets. The thickness of outgrown cell sheets varied in the presence of different types of culture media. MSC-PCM + 2% PRFext was the most effective medium for cell multilayer formation. Although cell growth in a horizontal plane is fundamentally different from cell growth in multiple layers, the observed effect was, to some extent, consistent with the growth rate results shown in Figure 3. By contrast, although calcium deposit formation largely relies on the nature of the original periosteum tissue segments, the conventional medium generally induced diffused mineralization, whereas MSC-PCM medium reduced it in limited regions.

Figure 6 shows the distribution of PDGF-B, transforming growth factor beta 1 (TGFβ1), and collagen type I in the outgrown cell sheets. PDGF-B or antigenically similar proteins were not detected in any groups in Figure 6. The expression of TGFβ1 or similar proteins was slightly positive in the periosteal sheets expanded in the conventional medium. However, collagen type I was detected in all groups. As MSC-PCM + 2% PRFext produced the thickest cell multilayers, the volume of collagen type I matrix was the most abundant in the periosteal sheets expanded in this culture medium.

Figure 7 shows the expression of the basic markers of mesenchymal stem cells, CD73, CD90, and CD105, in the cells growing in periosteal sheets. Comparison was performed only between two groups; namely, the conventional medium and MSC-PCM + 2% PRFext. Expression of CD105 was lower in the newly developed medium than that in the conventional medium; however, no statistical differences were observed.

## 3. Discussion

FBS is still considered a “magical” supplement for the successful cultivation of cells, although the associated disadvantages are well known. To improve the quality of the resulting cell-based products and their therapies, animal-derived factors should be completely eliminated from culture media. Several efforts have been directed toward the development of a chemically defined medium suitable for adherent cell cultures. In the initial phase of our project, we aimed to develop such a chemically defined medium or a medium free of animal components suitable for the cultivation of periosteal sheets. In comparison with single cell cultures, however, periosteal tissues require stronger adhesion systems that cannot be achieved by simply adding sufficient amounts of recombinant human adhesion molecules, such as fibronectin and vitronectin, as evident from our preliminary studies. Instead, such systems may be reproduced with the use of animal or human-derived sera. Therefore, we modified our aim to develop a xeno-free medium.

At the beginning of the second phase, we developed and patented a new expansion method using stocked human platelet-rich plasma (PRP) along with recombinant human bFGF to allow the growth of periosteal sheets [12]. This method provides a consistent source of fully confluent periosteal sheets in 100 mm dishes within 4 weeks. However, a thin fibrin membrane also forms, covering periosteal sheets that may cause easy detachment of periosteal sheets upon medium exchange. Thus, in the preliminary study, we attempted to evaluate alternative ways to utilize the factors from platelet concentrates.

In industry, it may be convenient and economical to use pooled allogeneic PRP, although complicated and costly extraction methods have to be introduced into the manufacturing process. For the preparation of small-scale homemade autologous PRP extracts, by contrast, the preparation protocol needs to be simple and cost-effective. The first choice is definitely platelet-rich fibrin (PRF) exudate or releasate. However, as various major adhesion molecules were found to be adsorbed on fibrin fibers in a preliminary experiment [Kawase et al., manuscript in submission], we homogenized the minced PRF preparations to release these adhesion molecules and used the obtained supernatant supplemented with small debris of fibrin fragments. This preparation protocol is fast, less labor-intensive, and produced better results during the initial adhesion and growth of periosteal sheets, even after reducing the content of PRFext to 2% (*v*/*v*).

The rapid growth induced by PRFext was not associated with the initial cell outgrowth, but was related to the acceleration of cell proliferation after outgrowth. As illustrated in Figure 8 and previously demonstrated [13], most periosteal cells are dead in the initial phase of culture, and the surviving cells actively replicate and migrate out to form periosteal sheets. Our results indicate that the added PRFext acted on cell outgrowth and subsequent cell proliferation, but not on cell turnover. The shortening of the cell turnover phase may allow further reduction in the period of periosteal sheet preparation to less than 3 weeks in the near future.

As rapid proliferation needs to be balanced against genetic, phenotypic, and functional stability [1], we examined the compatibility of periosteal sheets prepared using the new culture medium in the validation stage of this study. Regarding genetic stability, the source of periosteal sheets, i.e., the cells from alveolar periosteum, is at a relatively late stage of differentiation compared to mesenchymal stem cells. In general, the genetic instability of cells correlates with their pluripotency and multipotency [14,15]; therefore, the majority of periosteal cells may be relatively genetically stable during expansion. In support of this speculation, we have previously demonstrated the least probability of cell transformation in X-ray-irradiated periosteal cells [16]. Furthermore, the qualitative analysis of a limited number of cells in karyotype testing (preliminary study) revealed no abnormality in the chromosomes from periosteal sheet samples at the end of the expansion period.

Regarding the rest of the criteria, the type of culture medium failed to have any significant influence on the expression of the conventional surface markers of mesenchymal stem cells, i.e., CD73, CD90, and CD105 [17]. ALP expression and calcium phosphate deposition were, to some extent, influenced by culture media. The addition of PRFext suppressed the spontaneous increase in ALP activity and consequent calcium deposit formation observed in the periosteal sheets expanded in Medium199 + 10% FBS. By contrast, MSC-PCM increased the accumulation of collagen around periosteal cells and consequently increased the thickness of periosteal sheets with an increase in growth rate. MSC-PCM induced maximum effects on sheet thickness in combination with PRFext.

Similar observations were recorded in a previous study using another stem cell medium, MesenPRO-RS medium supplemented with 2% FBS [8]. Although the ALP activity and the ability to form calcium deposits in vitro were lower, the periosteal sheets prepared with this medium showed potent osteogenesis similar to that achieved with the conventional medium upon subcutaneous implantation in animal models. Taken together with the evidence that collagen provides a platform for mineral deposition [18], the periosteal sheets prepared with MSC-PCM + 2% PRFext may possibly exhibit compatible osteogenesis.

The shortening of the preparation period is beneficial for both clinics serving this regenerative therapy and patients receiving this therapy, in terms of cost, operation efficiency, and treatment schedule. However, compatibility must be predefined and tested to ensure safety and efficacy of the resulting periosteal sheets [1] prior to clinical application. As expected, the present study demonstrates that the critical qualities of the periosteal sheet prepared with MSC-PCM + 2% PRFext are not negatively influenced during the process of expansion. The process of blood collection from patients can be estimated to be relatively low on the basis of predicted consumption as mentioned: for a medium size (2−3 tooth width) alveolar ridge augmentation, approximately 30 periosteal sheets are usually prepared. When 60 mm culture dishes are used, approximately 600 mL of the culture medium and approximately 12 mL PRFext are required for the 4-week culture. Since a 10 mL whole-blood sample, including 1 mL Acid Citrate Dextrose Formula-A (ACD-A), produces approximately 2.5 mL PRFext, approximately 45 mL peripheral blood should be collected as a sufficient starting volume prior to the explant culture. However, in case of smaller bone defects, such as periodontal bone defect, the volume of blood required for the culture can be reduced to between one-fifth and one-tenth.

In addition, this xeno-free medium minimizes the risk of unknown pathogen contamination. The newly developed MSC-PCM medium is exceptionally more expensive than the conventional Medium199, but the total cost may be markedly reduced by choosing MSC-PCM + 2% PRFext. Therefore, we proposed that this complete xeno-free medium may serve as a promising replacement medium for the conventional FBS-containing medium in the preparation of periosteal sheets.

## 4. Materials and Methods

### 4.1. Preparation of PRFext

Blood was collected from six healthy and non-smoking volunteers aged 24–44 years (three females and three males) using butterfly needles (21G 3/400; NIPRO, Osaka, Japan) and Vacutainer tubes (Japan Becton, Dickinson and Company, Tokyo, Japan). To prepare the PRFext, the blood samples were immediately (within approximately 2 min from blood collection) centrifuged by a Medifuge centrifugation system (Silfradent S. r. l., Santa Sofia, Italy) [19,20]. The red thrombus (the fraction of red blood cells) was eliminated with scissors and the resulting PRF preparations were minced using scissors, followed by homogenization with sterile BioMasher (Nippi, Tokyo, Japan), as illustrated in Figure 9 and as described previously [21]. The homogenized samples were centrifuged at maximum speed to exclude fibrin matrix fragments. The resulting supernatant was stored at −80 °C until use. Approximately 2.5 mL PRFext can be prepared from 10 mL whole-blood sample, including 1 mL ACD-A. The levels of PDGF-BB in the resulting samples usually ranged from 25 to 50 ng/mL [21].

The study design and consent forms for all the procedures (project identification code: 2015-2143) were approved by the Ethics Committee for Human Subjects of the Niigata University School of Medicine (Niigata, Japan) on 12 June, 2017, in accordance with the Helsinki Declaration of 1964 as revised in 2013.

### 4.2. Explant Culture of Periosteum Tissue Segments to Form Periosteal Sheets

Six patients aged 20–44 years (four females and two males) in need of wisdom tooth extraction participated in this study after providing written informed consent. Aliquots of periosteum tissues were aseptically dissected from the buccal side of the retromolar region in the mandible of healthy donors, washed thrice in Dulbecco’s PBS without Ca^2+^ and Mg^2+^, cut into small segments (~1 × 1 mm), and plated on 60 mm dishes. After incubation for 15–20 min under dry conditions in a CO_2_ incubator, the conventional medium (Medium199 supplemented with 10% FBS), MSC-PCM medium (Kohjin Bio, Sakado, Japan) supplemented with 4% FBS, or MSC-PCM medium supplemented with 2% PRFext was added to cover the bottom surface of the dish. All media were commonly supplemented with 25 μg/mL of L-ascorbic acid, 100 U/mL of penicillin G, 100 μg/mL of streptomycin, and 0.25 μg/mL of amphotericin B (Invitrogen, Carlsbad, CA, USA). The volume of media was increased in a stepwise manner as cell outgrowth proceeded.

### 4.3. Evaluation of Cell Outgrowth and Growth Rate

The onset of cell outgrowth, which indicates days required for the migration of the first cell out of the original periosteum tissue segments, was evaluated using an inverted microscope once every 3 days. Frequent examination of cell outgrowth can sometimes lead to detachment of periosteum tissue segments; hence, we did not perform a daily evaluation.

The growth rate was determined by measuring the lengths of the long (major) axis and short (minor) axis. Periosteal sheets were plated on a light box and measured by a caliper.

### 4.4. Histological Determination of ALP Activity

For ALP staining, periosteal sheets were fixed with 10% neutralized formalin on dishes and directly treated with an ALP staining kit (Muto Chemicals, Tokyo, Japan) for 4 h, followed by counterstaining with Safranin-O [4].

### 4.5. Histological and Immunohistochemical Examination for Calcium Deposition, Growth Factor Expression, and Collagen Accumulation

Periosteal sheets were gently detached using a cell scraper and immediately fixed with 10% formaldehyde in 0.1 M phosphate buffer, pH 7.4, overnight. Fixed samples were dehydrated using an ethanol series (70%−100%) and xylene and embedded in paraffin. The samples were sagittally sectioned at a thickness of 6 μm [4]. 

As previously described [20], the deparaffinized sections were subjected to antigen retrieval with Liberate Antibody Binding Solution (Polysciences, Inc., Warrington, PA, USA) and blocked with Block-Ace (Sumitomo Dainippon Pharma., Osaka, Japan) solution in 0.1% Tween-20-containing PBS. The sections were probed with a rabbit polyclonal anti-collagen type I antibody (1:400; Bioss Inc., Boston, MA, USA), anti-TGFβ1 antibody (1:200; Santa Cruz Biotechnology, Inc., Santa Cruz, CA, USA), or anti-PDGF-B (1:200; Santa Cruz) diluted in ImmunoShot Mild (Cosmo Bio, Tokyo, Japan) overnight at 4 °C, followed by incubation in horseradish peroxidase (HRP)-conjugated anti-rabbit IgG (Cell Signaling Technology, Danvers, MA, USA). Immunoreactive proteins were visualized with a 3,3′-diaminobenzidine (DAB) substrate solution (Kirkegaard & Perry Laboratories, Inc., Gaithersburg, MD, USA).

The sections were alternatively stained with hematoxylin and eosin (HE) or silver nitrate (von Kossa staining). For von Kossa staining, the sections were faintly counterstained with Kernechtrot solution to provide a background stain [4].

### 4.6. Flow Cytometric Evaluation of CD73-, CD90-, and CD105-Positive Periosteal Cells

Cells were dispersed from cultured periosteal sheets with 0.05% trypsin + 0.53 mM ethylenediaminetetraacetic acid (EDTA) solution (Invitrogen), washed twice with PBS, and suspended in 0.1 mL PBS containing 0.1% bovine serum albumin (BSA) at a density of 1 × 10^6^ cells/mL. The cells were probed for 30 min at 4 °C with 5 μL of the following mouse monoclonal antibodies: anti-CD73-fluorescein isothiocyanate (FITC) (IgG1) (BioLegend, San Diego, CA, USA), anti-CD90-phycoerythrin (PE)/Cy5 (IgG1) (BioLegend), and anti-CD105-PE (IgG1) (BioLegend). After being washed twice with PBS, the cells were analyzed by flow cytometry (Navios; Beckman Coulter, Miami, FL, USA). Data analysis and histogram overlay were performed using Navios Software (Beckman Coulter). For isotype controls, individual corresponding antibodies (all from BioLegend) were used [8].

### 4.7. Statistical Analysis

The data were expressed as mean ± standard deviation (SD). For multigroup comparisons, statistical analyses were performed to compare the mean values by Kruskal–Wallis one-way analysis of variance, followed by Steel–Dwass multiple comparison test (BellCurve for Excel version 3.00; Social Survey Research Information Co., Ltd., Tokyo, Japan). For two group comparisons, statistical differences were tested using the Mann-Whitney rank-sum test (SigmaPlot 12.5; Systat Software, Inc., San Jose, CA, USA). Differences with P values of less than 0.05 were considered statistically significant.

## 5. Conclusions

This preclinical study successfully validated the applicability of PRFext for FBS replacement in explant cultures of periosteum tissue segments to form periosteal sheets. These findings were established using the samples donated by healthy volunteers. For therapeutic use, however, autologous periosteum tissue and PRFext must be employed for the preparation of periosteal sheets. Thus, the efficacy of PRFext and the responsiveness of periosteum tissue may vary with individual samples, and more careful measures should be adopted while evaluating the quality of periosteal sheets prepared by this protocol.

## Figures and Tables

**Figure 1 ijms-20-01053-f001:**
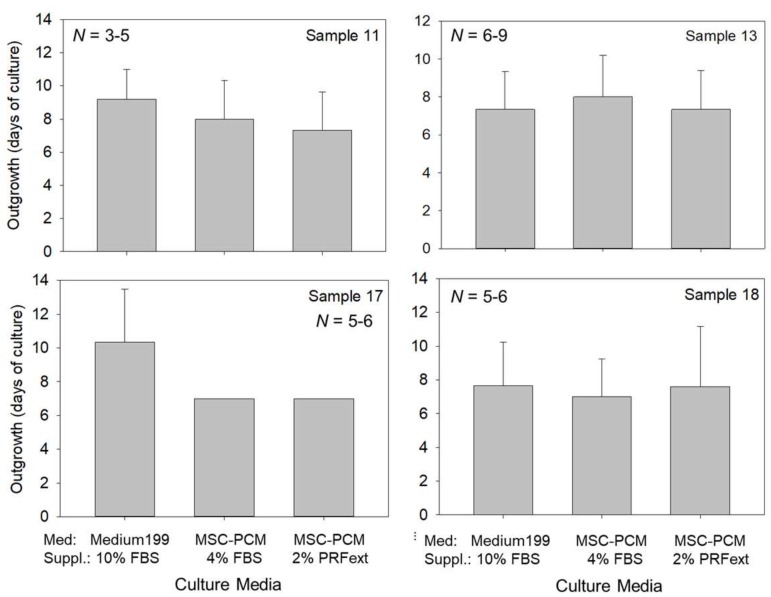
Effects of different culture media on the onset of periosteal cell outgrowth. The data obtained from periosteum samples derived from four independent donors are shown. *X*-axis: types of culture media. Statistical analysis was performed by Kruskal–Wallis one-way analysis of variance, followed by Steel–Dwass multiple comparison test. No significant difference was observed between the groups. *N* = 3, 4, 5, 6, or 9 replicates.

**Figure 2 ijms-20-01053-f002:**
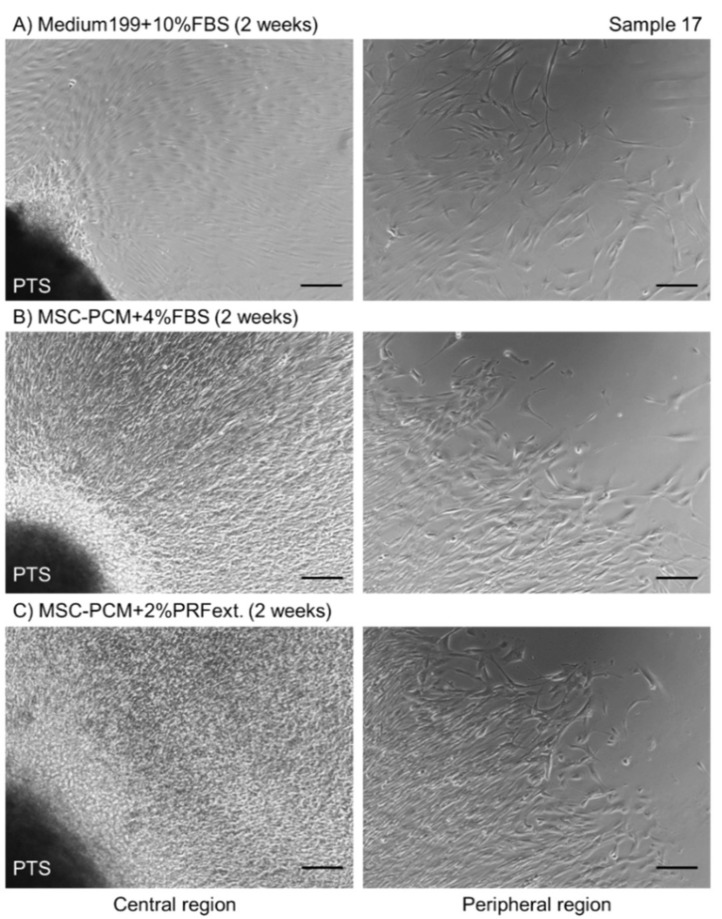
Photomicrographs of periosteal cells in the central and peripheral regions of periosteal sheets cultured in different culture media. (**A**) Medium199 + 10% fetal bovine serum (FBS), (**B**) MSC-PCM + 4% FBS, (**C**) MSC-PCN + 2% platelet-rich fibrin extract (PRFext). Bar = 50 µm. PTS: periosteum tissue segment.

**Figure 3 ijms-20-01053-f003:**
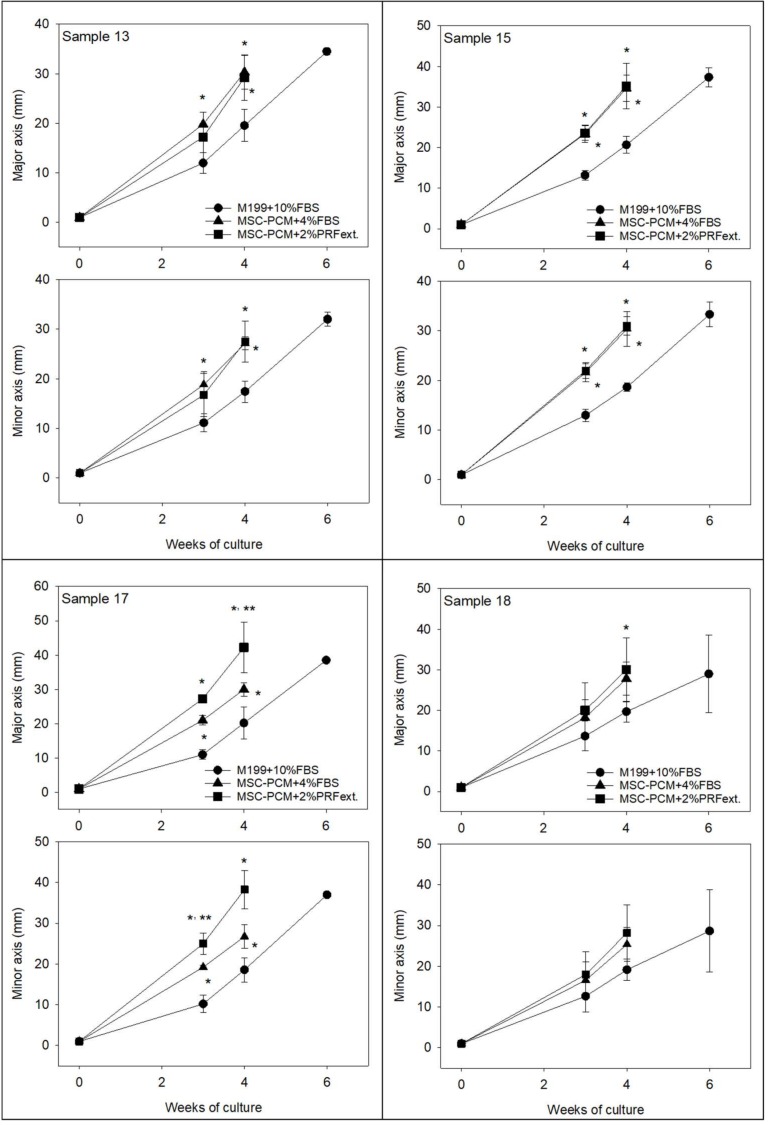
Effects of different culture media on the growth of periosteal sheets. The data obtained from periosteum samples derived from four independent donors are shown. *X*-axis: time periods (weeks) of explant culture. *N* = 2 (6 weeks), 3 (6 weeks), 4, 5, or 7 replicates. Statistical analysis was performed by Kruskal–Wallis one-way analysis of variance, followed by Steel–Dwass multiple comparison test. * *p* < 0.05 as compared with the control group (Medium199 + 10% FBS) at same time points. ** *p* < 0.05 as compared with the other experimental group (MSC-PCM + 4% FBS) at same time points.

**Figure 4 ijms-20-01053-f004:**
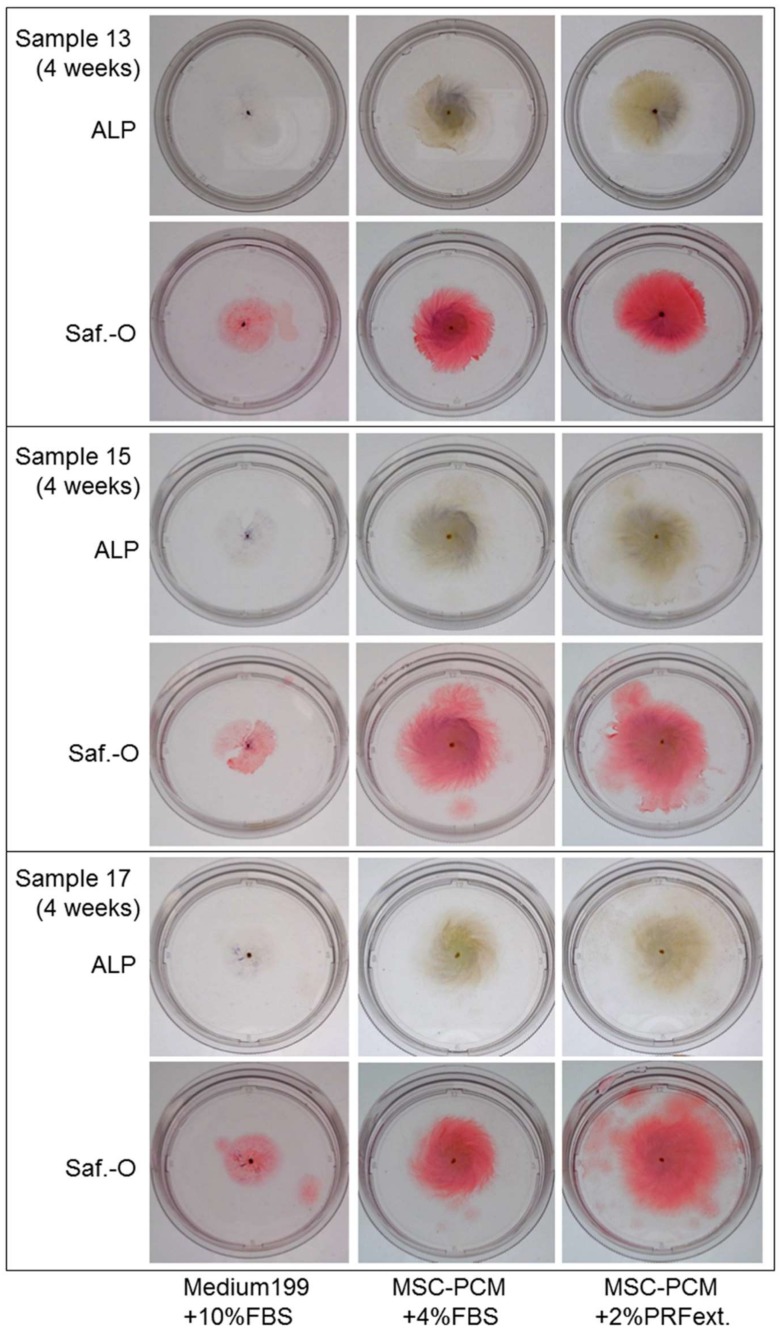
Effects of different culture media on the alkaline phosphatase (ALP) activity and size of periosteal sheets. Fixed individual periosteal sheets were first stained for ALP activity (positive: dark blue-purple) and subsequently treated with Safranin-O (Saf.-O) for the evaluation of their sizes. We used 60 mm culture dishes.

**Figure 5 ijms-20-01053-f005:**
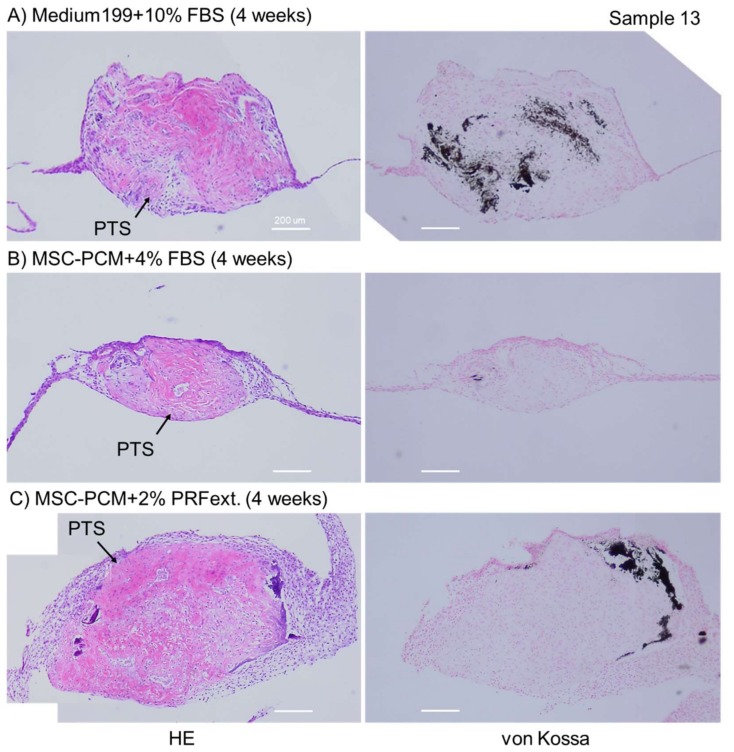
Effects of different culture media on the thickness of periosteal sheets. In von Kossa staining, calcium deposits were stained black. These data are representative of five independent experiments. hematoxylin and eosin (HE) staining. Bar = 200 µm.

**Figure 6 ijms-20-01053-f006:**
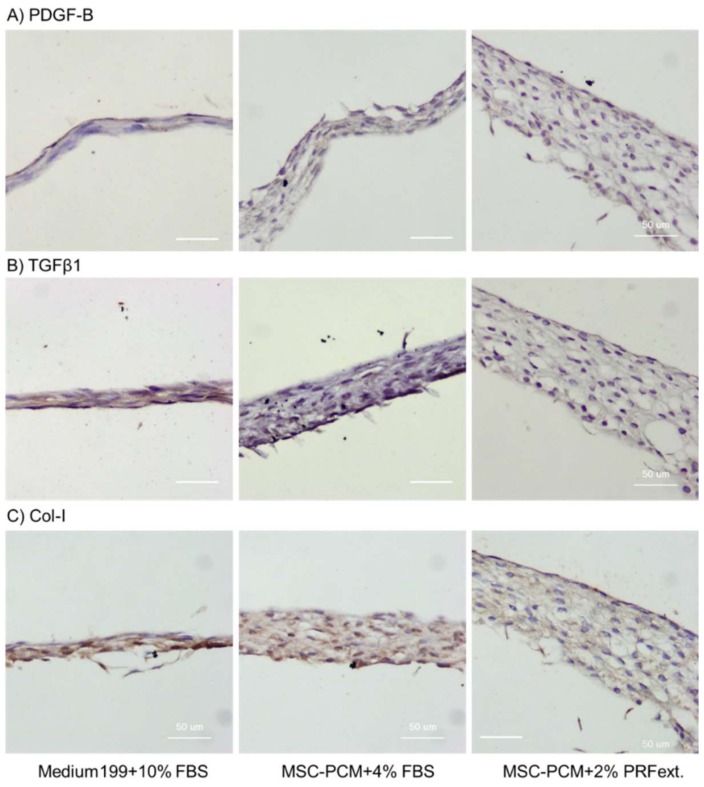
Effects of different culture media on the expression of platelet-derived growth factor-B (PDGF-B), transforming growth factor beta 1 (TGFβ1), and collagen type I in the central region of periosteal sheets (outgrowth area). Immunohistochemical staining with visualization using 3′-diaminobenzidine (DAB) (positive: dark brown). These data are representative of five independent experiments. Bar = 50 µm.

**Figure 7 ijms-20-01053-f007:**
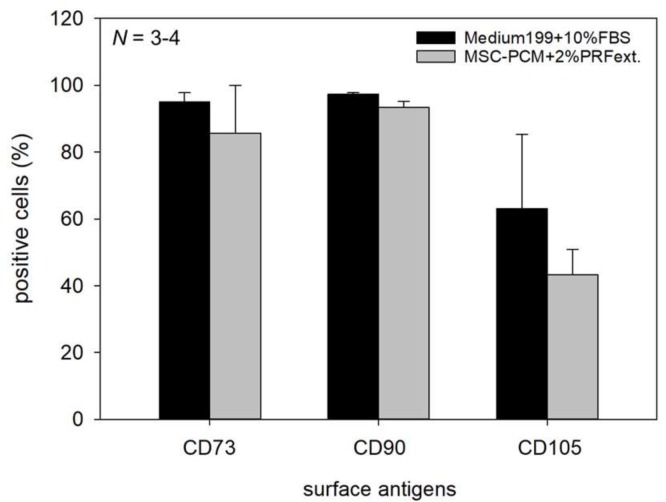
Effects of different culture media on the expression of surface antigens. Only periosteal sheets cultured with MSC-PCM + 2% PRFext were compared with the control sheets (Sample 21). *X*-axis: type of surface antigens. Statistical analysis was performed using the Mann–Whitney rank-sum test and no significant difference was observed between the two groups. *N* = 3 or 4 replicates.

**Figure 8 ijms-20-01053-f008:**
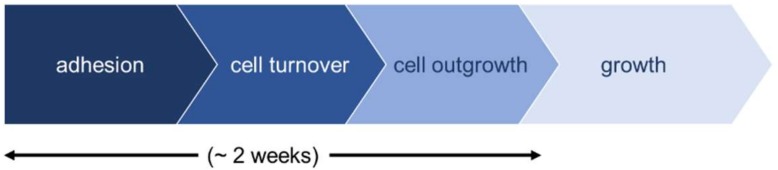
Phases in the process of periosteal sheet cultures.

**Figure 9 ijms-20-01053-f009:**
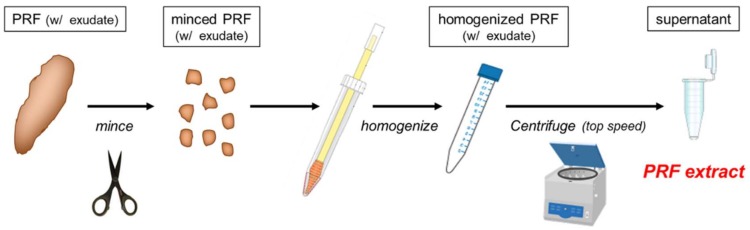
Graphic summary of preparation of platelet-rich fibrin (PRF) extract.

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
