# Peer review of "Platelet-Rich Fibrin Extract: A Promising Fetal Bovine Serum Alternative in Explant Cultures of Human Periosteal Sheets for Regenerative Therapy"

_ijms, 2019, doi:10.3390/ijms20051053_

Round 1
Reviewer 1 Report
This manuscript describes the characterization of platelet rich fibrin extract PRF -ext as replacement for fetal bovine serum (FBS) in expansion cultures for human periosteal sheets.
In general, defined culture media are warranted for clinical application of cultured human cells, so the authors are correct in trying to achieve this goal by replacing FBS by more defined components.
This paper describes some comparative studies between culture media with either 10% FBS, 4% FBS or 2% PRF-ext. Most of the experiments are relevant, but the methodology and robustness of the experiments is not described in enough detail for the reader to follow the reasoning and conclusions.
Methodology
Please describe in the legend for each figure what is depicted on x-axis as well as Y-axis.
Figure 1: it is unclear what is meant by ‘outgrowth (days of culture)' . In materials and methods it is stated that outgrowth was measured by the length of long axis and short axis. This is not indicated in Figure 1, whereas in Figure 3 major axis and minor axis are mentioned. The number of samples and number of independent donors are mixed up; in general N is used for number of independent samples, and these can be measured or evaluated with a certain number of dependent observations in a single experiments.
Figure 3: in the text it is stated that 2% PRFext was most effective, however, these samples showed equal growth curves to 4% FBS for many samples/conditions. Furthermore, it would seem to me that an analysis over a mean of all samples would be more relevant and give statistically stronger data. Why were these analyses performed in a per sample fashion?
Figure 4: if the groups cannot be compared, it is of little value to show these data.
Figure 5 shows differences in cell multilayers, which the authors state to be consistent with fig 3. However, in figure 3 there was in many cases no difference between 4% FBS and 2% PRFext.
Figure 6 shows marked differences between the culture conditions . Why not quantify these data, especially sheet thickness?
Figure 7. was this done with only one sample?
Statistics: with such a limited number of samples a non-parametric test should be performed rather than Student t test (which should be used on normal distribution of data, this cannot be assumed with such a low N of 6 volunteers).
Discussion
Please comment of the volume of blood necessary to prepare the PRF ext? What is the implication for clinical applicability?
Author Response
This paper describes some comparative studies between culture media with either 10% FBS, 4% FBS or 2% PRF-ext. Most of the experiments are relevant, but the methodology and robustness of the experiments is not described in enough detail for the reader to follow the reasoning and conclusions.
Response: According to the other reviewers’ comments, additional notes were inserted into the materials and methods section for improved clarity.
Methodology
Please describe in the legend for each figure what is depicted on x-axis as well as Y-axis.
Response: We have added short notes for X-axis in the selected figure legends.
Figure 1: it is unclear what is meant by ‘outgrowth (days of culture)' . In materials and methods it is stated that outgrowth was measured by the length of long axis and short axis. This is not indicated in Figure 1, whereas in Figure 3 major axis and minor axis are mentioned. The number of samples and number of independent donors are mixed up; in general N is used for number of independent samples, and these can be measured or evaluated with a certain number of dependent observations in a single experiments.
Response: The “onset of cell outgrowth” we used in this study indicates days required for the migration of the first cell out of the original periosteum tissue segments (please see Figure 8). This short explanation was added to the results section and also to the materials and methods section.
As for the number of samples and the number of independent donors, we agree with your interpretation. However, cell outgrowth and cell growth in expansion cultures of periosteum tissue segments are highly specific for individual segments. Thus, cell outgrowth is influenced by the number/density of cells contained in periosteum tissue segments and adhesive potential of periosteum tissue segments to plastic dishes (and also operators’ skills) more than the nature of individual samples. If sufficient adhesion is not acquired, cell outgrowth is severely suppressed regardless of cell density in segments.
When periosteum tissue is cut into 10 segments, because of non-uniform cell distribution in the individual divided segments, the data obtained from the resulting periosteal sheets can hardly be expressed “in replicates of ten.” In our opinion, these segments should be considered as an “independent sample.” This explanation was added to the materials and methods section.
On the other hand, it is generally accepted that the growth activity or replication of isolated cells, e.g., bone marrow stem cells, is significantly influenced by age, gender or other factors related to individuals. However, we have observed in more than 300 periosteum samples that periosteal cell outgrowth is not reproducibly influenced by these factors. This effect can be confirmed by the data of individual donors in Figure 1. Typically, periosteal cell outgrowth is usually observed between 7 to 12 days of culture in the conventional medium. The necessity of this interval for cell outgrowth may be understood by considering the mechanism behind the cell outgrowth (reference #10 below). For your reference, further information regarding the basic characteristics of periosteal sheets and tips for their handling is mentioned in more detail in the following articles.
1) Kawase T*, Okuda K, Nagata M, Yoshie H. [Chapter 2] The cell-multilayered periosteal sheet: a promising osteogenic and osteoinductive grafting material. In: “New Trends in Tissue Engineering and Regenerative Medicine” Ed by Ueda M. InTech Open Access Publisher (Rijeka, Croatia), pp19-35, 2014.
2) Uematsu K, Nagata M*, Kawase T, Suzuki K, Takagi R. Application of stem cell media to explant culture of human periosteum: an optimal approach for preparing osteogenic cell material. J Tissue Eng 4:2041731413509646; 2013.
3) Kawase T*, Uematsu K, Nagata M, Okuda K, Burns DM, Yoshie H. (Proceeding) Biological and biomechanical characterization of highly self-multilayered human periosteal sheets as an osteogenic grafting material. Cytotherapy 15(4) (supplement):S45-46; 2013.
4) Horimizu M, Kawase T*, Tanaka T, Okuda K, Nagata M, Burns DM, Yoshie H. Biomechanical evaluation by AFM of cultured human cell-multilayered periosteal sheets. Micron 48:1-10; 2013.
5) Uematsu K, Kawase T*, Nagata M, Suzuki K, Okuda K, Yoshie H, Burns DM, Takagi R. Tissue culture of human alveolar periosteal sheets using a stem-cell culture medium (MesenPRO-RSTM): In vitro expansion of CD146-positive cells and concomitant upregulation of osteogenic potential in vivo. Stem Cell Res 10(1):1-19; 2013.
6) Kawase T, Nakata K. [Chapter 7] Preservation and maintenance of periosteal sheets. In “Clinical study and development in regenerative medicine” Technical Information Institute Co, Ltd. (Tokyo, Japan), pp148-152, 2013. (in Japanese)
7) Kawase T*, Tanaka T, Nishimoto T, Okuda K, Nagata M, Burns DM, Yoshie H. An osteogenic grafting complex composed of human periosteal sheet and a porous poly(L-lactic acid) membrane scaffold: Biocompatibility, biodegradability, and cell fate in vivo. J Bioact Compat Polym 27(2):107-121; 2012.
8) Kawase T*, Okuda K, Yoshie H. Biomaterials for periodontal tissue regenerative therapy; Characteristics of human autologous periosteal sheet. Microscopy 47(4):216-222; 2012. (in Japanese)
9) Kawase T*, Okuda K, Yoshie H. Cultured periosteal sheets. In “Oral and dental fields” of “Monographs of regenerative medicine” Asakura Publishing Co., Ltd. (Tokyo, Japan), pp53-67, 2012. (in Japanese)
10) Kawase T*, Kogami H, Nagata M, Uematsu K, Okuda K, Burns DM, Yoshie H. Manual cryopreservation of human alveolar periosteal tissue segments: Effects of pre-culture on recovery rate. Cryobiol 62(3):202-209; 2011.
11) Kawase T*, Tanaka T, Nishimoto T, Okuda K, Nagata M, Burns DM, Yoshie H. Improved adhesion of human cultured periosteal sheets to a porous poly(L-lactic acid) membrane scaffold without the aid of exogenous adhesion biomolecules. J Biomed Mater Res A 98(7):100-113; 2011.
12) Kawase T*, Yamanaka K, Suda Y, Kaneko T, Okuda K, Kogami H, Nakayama H, Nagata M, Wolff LF, Yoshie H. Collagen-coated poly(L-lactide-co-ε-caprolactone) film: A promising scaffold for cultured periosteal sheets. J Periodontol 81(11):1653-1662; 2010.
13) Kawase T*, Okuda K, Kogami H, Nakayama H, Nagata M, Yoshie H. Osteogenic activity of human periosteal sheets cultured on salmon collagen-coated ePTFE meshes. J Mater Sci Mater Med 21(2):731-739; 2010.
14) Kawase T*, Okuda K, Kogami H, Nakayama H, Nagata M, Nakata K, Yoshie H. Characterization of human cultured periosteal sheets expressing bone-forming potential: in vitro and in vivo animal studies. J Tissue Eng Reg Med 3:218-229; 2009.
Thus, in this study, we basically collected and analyzed data obtained from segments derived from the same donors in most figures. However, the only exception is in Figure 6, where we have mixed up the data obtained from the different donors (samples).
Figure 3: in the text it is stated that 2% PRFext was most effective, however, these samples showed equal growth curves to 4% FBS for many samples/conditions. Furthermore, it would seem to me that an analysis over a mean of all samples would be more relevant and give statistically stronger data. Why were these analyses performed in a per sample fashion?
Response: As you indicated, in some samples, 2% PRFext was almost as potent as 4% FBS in terms of cell growth in a horizontal plane. However, we should recognize that in some cases, 2% PRFext was more potent than 4% FBS but 4% FBS was never more potent than 2% PRFext. Therefore, the order of potency could be evaluated roughly as 2% PRFext (MSC-PCM) ≥ 4% FBS (MSC-PCM) >> 10% FBS (the conventional medium). According to your comment, we have modified the expression regarding the order of potency in the text.
As per your suggestion regarding a mean of all samples, we think it is a good suggestion for better contrast. However, our aim here is to show that the growth of periosteal sheets does not vary with individual donors, at least when cultured with the conventional medium. In contrast to bone marrow stem cells, this is one of the important characteristics of periosteum expansion culture. As far as we have tested in more than 300 donors, neither age nor gender of donors is a critical factor to influence the fate and potency of periosteum tissue.
Figure 4: if the groups cannot be compared, it is of little value to show these data.
Response: Because it was difficult to quantify the percentage of ALP-positive cells and the levels of ALP activity in these images, we have presented only qualitative data. However, ALP activity is a very important marker of matured osteoblasts. Along with the histological data for mineral deposit formation by von Kossa staining (Fig. 5), we have also evaluated the maturation stages of periosteal cells expressing osteoblastic phenotype in the growing periosteal sheets. Due to poor quality of macroscopic photographs (Fig. 4), we are doubtful that all readers may not be able to clearly understand the effects of culture media on the osteoblastic differentiation in the expansion culture. Thus, we believe Figure 4 is critical for better understanding of the article.
Figure 5 shows differences in cell multilayers, which the authors state to be consistent with fig 3. However, in figure 3 there was in many cases no difference between 4% FBS and 2% PRFext.
Response: Strictly speaking, cell growth in a horizontal plane is distinguished from cell growth in multiple layers. This is because cell accumulation in multiple layers needs preceding deposition of extracellular matrix (ECM) as a cell-scaffolding material. Cell growth activity and ECM production activity are not necessarily associated with each other. In the results section, we explained this possibility and modified the expression.
Figure 6 shows marked differences between the culture conditions. Why not quantify these data, especially sheet thickness?
Response: As for PDGF-B and TGFβ1, it is theoretically possible to stain both periosteal cells and extracellular matrix. However, in this study, expanded regions of periosteal sheets were only faintly positive and they were too weak to be quantified by staining. As for collagen type-I, it was diffusely stained all over in the extracellular matrix, which made it difficult to quantify it.
Also, for sheet thickness, it is difficult to perform quantification of sheet thickness, though it is easy to do so in images. Because this thickness of cell-layers are relatively constant, it is easy to measure the thickness of thin cell-layers. In contrast, because relatively thick cell-layers cannot be enlarged without decreases in the thickness, an accurate estimate of their thickness cannot be made. The thickness decreases as the distance from periosteum tissue segments increases. In addition, it is practically difficult to sagittally section periosteal sheets precisely along a vertical plane. Thus, we can roughly measure their thickness from the sagittal sections, but its quantification is not accurate enough for comparison.
Figure 7. was this done with only one sample?
Statistics: with such a limited number of samples a non-parametric test should be performed rather than Student t test (which should be used on normal distribution of data, this cannot be assumed with such a low N of 6 volunteers).
Response: Yes, these data were obtained from the experiment using the sample obtained from a single donor. We could use 3 periosteal sheets (Medium 199 + 10% FBS) and 4 periosteal sheets (MSC-PCM + 2% PRFext) for FCM analysis only in Sample # 21. As for other donors, the number of periosteal sheets preliminarily used for FCM analysis was limited only to 2. Because those data were essentially similar, we decided to present the data obtained from samples of the single donor in Figure 7.
Thank you for your advice on statistics. We performed the Mann-Whitney Rank Sum Test (SigmaPlot) to test statistical differences in Figure 7. The results were “not statistically different” in any comparisons. This was described in the materials and methods section.
Discussion
Please comment of the volume of blood necessary to prepare the PRFext? What is the implication for clinical applicability?
Response: We have added the estimated volume of blood samples required for the 4-week expansion culture. In case of the medium size (2−3 tooth width) of alveolar ridge augmentation, approximately 45 mL blood is required to prepare 12 mL PRFext, which is sufficient for a culture of 30 dishes (diameter = 60 mm), including several reserves.
According to the approved protocol, we implanted the periosteal sheets along with PRP and crushed bone (or bone substitutes) in clinical setting. At present, it is difficult to predict what will be the clinical outcomes of the improved culture protocol. However, at least, we can emphasize that the reduction in the preparation time is beneficial for both, clinics serving this regenerative therapy and patients receiving this therapy in terms of cost, operation efficiency, and treatment schedule. However, compatibility must be predefined and tested to ensure safety and efficacy of the resulting periosteal sheets prior to any clinical application. Thus, to meet this necessity, we performed this study.
Reviewer 2 Report
My opinion on this is that the introduction is well written and the scientific question is pertinent and the study is well designed. The main problem with this study is the number of cases analyzed. They should be at least 5 patients for each group. Furthermore, it is not clear in the figures to what the number "N = 3-5" and similar is referred to. Are 3 or 5 replicates? It should be specified. Furthermore the statistic is not appropriate for the number of samples analyzed. It would take non-paramentum tests that take into account the number of relics for each sample and the small number.
Author Response
The main problem with this study is the number of cases analyzed. They should be at least 5 patients for each group. Furthermore, it is not clear in the figures to what the number "N = 3-5" and similar is referred to. Are 3 or 5 replicates? It should be specified.
Response: In this study, we routinely divided the obtained piece of periosteum tissue (~5 × ~5 mm) into 20 segments or more, in 3 groups (≥6 segments per group). However, owing to their weak adhesion or poor cell distribution, cell outgrowth was not always observed for all the segments. In addition, some segments were detached and did not reattach. Thus, number of segments (N) was sometimes less than 6 per group. Owing to this problem, we were unsure if “N” can be considered as the number of replicates in this case. However, to avoid readers’ confusion, we specified it as “replicate” in the revised munscript.
In most figures, we analyzed the data in individual samples (independent donors). The purpose of this way of presentation is to demonstrate as to what extent are individual samples different from one another, giving more information to the readers who believe that data are largely influenced by individual differences. The data showed considerable variability with cell distribution, ECM deposition and handling skills rather than factors related to individual donors.
Furthermore the statistic is not appropriate for the number of samples analyzed. It would take non-paramentum tests that take into account the number of relics for each sample and the small number.
Response: We are afraid that we do not correctly understand what you mean by the above-mentioned comment. However, according to the comment given by Reviewer 1, we again tested the statistical differences by the Mann-Whitney Rank Sum Test (SigmaPlot) in Figure 7. For multiple comparisons in Figures 1 and 3, we performed Kruskal-Wallis one-way analysis of variance and subsequently Steel-Dwass multiple comparison test to confirm statistical differences. We hope that you will be find this this revision acceptable.
Round 2
Reviewer 1 Report
Most points have been clarified or improved by the authors.
Some minor comments remain:
Abstract: the claim ‘Periosteal sheets grew faster and thicker…’: no quantitative data on thickness were analysed, so there is no statistically significant data to substantiate this claim. Please reword.
Introduction: page 2, line 11: please correct: as well as osteoinducibility.
Author Response
Abstract: the claim ‘Periosteal sheets grew faster and thicker…’: no quantitative data on thickness were analysed, so there is no statistically significant data to substantiate this claim. Please reword.
Response: We deleted “and thicker.”
Introduction: page 2, line 11: please correct: as well as osteoinducibility.
Response: We corrected this word to “as well as osteoinductivity.”
Reviewer 2 Report
17 out of 28 references pertain to the author's group, please reduce the number of autocitations.
Author Response
17 out of 28 references pertain to the author's group, please reduce the number of autocitations.
Response: According to your suggestion, we reduced autocitations from 17 to 10.